# Iterated learning for emergent systematicity in VQA

**Ankit Vani**[*]
Mila, Université de Montréal

**Max Schwarzer**
Mila, Université de Montréal

**Yuchen Lu**
Mila, Université de Montréal

**Eeshan Dhekane**
Mila, Université de Montréal

**Aaron Courville**
Mila, Université de Montréal, CIFAR Fellow

## Abstract

Although neural module networks have an architectural bias towards compositionality, they require gold standard layouts to generalize systematically in practice. When instead learning layouts and modules jointly, compositionality does not arise automatically and an explicit pressure is necessary for the emergence of layouts exhibiting the right structure. We propose to address this problem using iterated learning, a cognitive science theory of the emergence of compositional languages in nature that has primarily been applied to simple referential games in machine learning. Considering the layouts of module networks as samples from an emergent language, we use iterated learning to encourage the development of structure within this language. We show that the resulting layouts support systematic generalization in neural agents solving the more complex task of visual question-answering. Our regularized iterated learning method can outperform baselines without iterated learning on SHAPES-SyGeT (**SHAPES Sy**stematic **Ge**neralization **T**est), a new split of the SHAPES dataset we introduce to evaluate systematic generalization, and on CLOSURE, an extension of CLEVR also designed to test systematic generalization. We demonstrate superior performance in recovering ground-truth compositional program structure with limited supervision on both SHAPES-SyGeT and CLEVR.

## 1 Introduction

Although great progress has been made in visual question-answering (VQA), recent methods still struggle to generalize systematically to inputs coming from a distribution different from that seen during training (Bahdanau et al., 2019b;a). Neural module networks (NMNs) present a natural solution to improve generalization in VQA, using a symbolic *layout* or *program* to arrange neural computational modules into computation graphs. If these modules are learned to be specialized, they can be composed in arbitrary legal layouts to produce different processing flows. However, for modules to learn specialized roles, programs must support this type of compositionality; if programs reuse modules in non-compositional ways, modules are unlikely to become layout-invariant.

This poses a substantial challenge for the training of NMNs. Although Bahdanau et al. (2019b) and Bahdanau et al. (2019a) both observe that NMNs can systematically generalize if given human-designed ground-truth programs, creating these programs imposes substantial practical costs. It becomes natural to jointly learn a *program generator* alongside the modules (Johnson et al., 2017b; Hu et al., 2017; Vedantam et al., 2019), but the generated programs often fail to generalize systematically and lead to worse performance (Bahdanau et al., 2019b).

*Iterated learning* (IL) offers one way to address this problem. Originating in cognitive science, IL explains how language evolves to become more compositional and easier-to-acquire in a repeated transmission process, where each new generation acquires the previous generation's language through a limited number of samples (Kirby et al., 2014). Early works with human participants (Kirby et al., 2008) as well as agent-based simulations (Zuidema, 2003) support this hypoth-

---

[*]Correspondance at: `ankit.vani@umontreal.ca`.

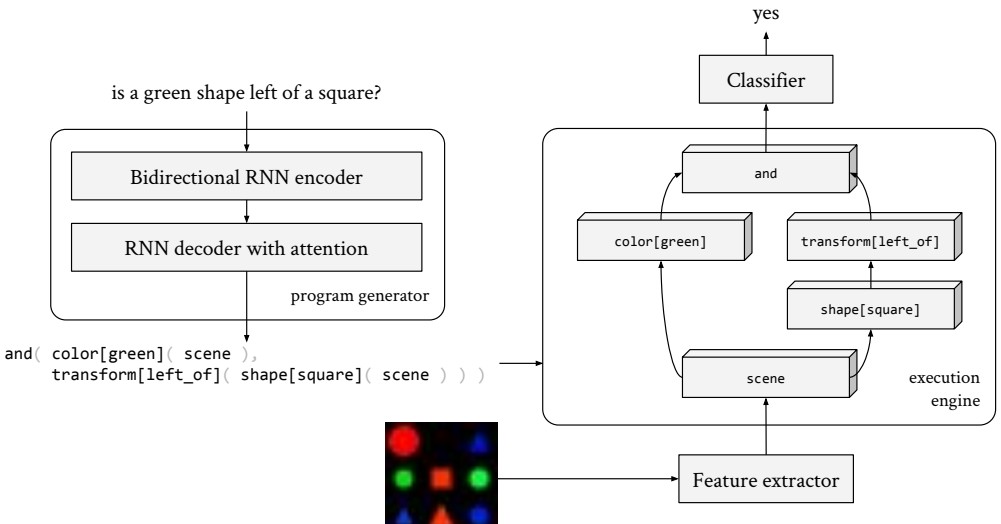

Figure 1: An overview of neural module networks (NMNs). A question $q$ is read by the *program generator* to produce a program $\hat{z}$. The *execution engine* assembles neural modules according to the layout $\hat{z}$ and feeds the input image $x$ into the assembled module network. A classifier takes the output of the top-level module to produce an answer $\hat{y}$ for the given $(q, x)$ pair.

esis. The machine learning community has also recently shown an increasing interest in applying IL towards emergent communication (Guo et al., 2019; Li & Bowling, 2019; Cogswell et al., 2019; Dagan et al., 2020; Ren et al., 2020). Different from previous works, we believe that IL is an algorithmic principle that is equally applicable to recovering compositional structure in more general tasks. We thus propose treating NMN programs as samples from a "layout language" and applying IL to the challenging problem of VQA. Our efforts highlight the potential of IL for broader machine learning applications beyond the previously-explored scope of language emergence and preservation (Lu et al., 2020).

To demonstrate our method, we introduce a lightweight benchmark for systematic generalization research based on the popular SHAPES dataset (Andreas et al., 2016) called SHAPES-SyGeT (**SHAPES Sy**stematic **Ge**neralization **T**est). Our experiments on SHAPES-SyGeT, CLEVR (Johnson et al., 2017a), and CLOSURE (Bahdanau et al., 2019a) show that our IL algorithm accelerates the learning of compositional program structure, leading to better generalization to both unseen questions from the training question templates and unseen question templates. Using only 100 ground-truth programs for supervision, our method achieves CLEVR performance comparable to Johnson et al. (2017b) and Vedantam et al. (2019), which use 18000 and 1000 programs for supervision respectively.

## 2 RELATED WORK

**Systematic generalization.** Systematicity was first proposed as a topic of research in neural networks by Fodor & Pylyshyn (1988), who argue that cognitive capabilities exhibit certain symmetries, and that representations of mental states have combinatorial syntactic and semantic structure. Whether or not neural networks can exhibit systematic compositionality has been a subject of much debate in the research community (Fodor & Pylyshyn, 1988; Christiansen & Chater, 1994; Marcus, 1998; Phillips, 1998; Chang, 2002; Marcus, 2018; van der Velde et al., 2004; Botvinick & Plaut, 2009; Bowers et al., 2009; Brakel & Frank, 2009; Fodor & Lepore, 2002; Marcus, 2018; Calvo & Symons, 2014).

Bahdanau et al. (2019b) investigate various VQA architectures such as neural module networks (NMNs) (Andreas et al., 2016), MAC (Hudson & Manning, 2018), FiLM (Perez et al., 2018), and relation networks (Santoro et al., 2017) on their ability to systematically generalize on a new synthetic dataset called SQOOP. They show that only NMNs are able to robustly solve test problems, but succeed only when a fixed tree-structured layout is provided. When learning to infer the module

network layout, robust tree-structured layouts only emerged if given a strong prior to do so. The authors conclude that explicit regularization and stronger priors are required for the development of the right layout structure.

CLEVR (Johnson et al., 2017a) is a popular VQA dataset, and various benchmarks achieve almost-perfect CLEVR validation scores (Hu et al., 2017; Hudson & Manning, 2018; Perez et al., 2018; Santoro et al., 2017). Bahdanau et al. (2019a) proposed an extension of CLEVR with a new evaluation dataset called CLOSURE, containing novel combinations of linguistic concepts found in CLEVR. The authors found that many of the existing models in the literature fail to systematically generalize to CLOSURE. Moreover, there is a significant gap between the performance achieved with ground-truth layouts and learned layouts on CLOSURE.

**Language emergence and compositionality.** Agents interacting in a cooperative environment can learn a language to communicate to solve a particular task. The emergence of such a communication protocol has been studied extensively in multi-agent referential games. In these games, one agent must describe what it saw to another agent, which is tasked with figuring out what the first agent saw (Lewis, 2008; Skyrms, 2010; Steels & Loetzsch, 2012). To encourage a dialogue between agents, several multi-stage variants of such a game have also been proposed (Kottur et al., 2017; Evtimova et al., 2018). Most approaches to learning a discrete communication protocol between agents use reinforcement learning (Foerster et al., 2016; Lazaridou et al., 2017; Kottur et al., 2017; Jorge et al., 2016; Havrylov & Titov, 2017). However, the Gumbel straight-through estimator (Jang et al., 2017) can also be used (Havrylov & Titov, 2017), as can backpropagation when the language in question is continuous (Foerster et al., 2016; Sukhbaatar & Fergus, 2016; Singh et al., 2019).

Several works in the literature have found that compositionality only arises in emergent languages if appropriate environmental pressures are present (Kottur et al., 2017; Choi et al., 2018; Lazaridou et al., 2018; Chaabouni et al., 2020). While generalization pressure is not sufficient to guarantee compositionality, compositional languages tend to exhibit better systematic generalization (Bahdanau et al., 2019b; Chaabouni et al., 2020). The community still lacks strong research indicating what general conditions are necessary or sufficient for compositional language emergence.

**Iterated learning.** The origins of the compositionality of human language, which leads to an astounding open-ended expressive power, have attracted much interest over the years. Kirby (2001) suggests that this phenomenon is a result of a *learning bottleneck* arising from the need to learn a highly expressive language with only a limited set of supervised learning data. The iterated application of this bottleneck, as instantiated by IL, has been demonstrated to cause artificial languages to develop structure in experiments with human participants (Kirby et al., 2008; Silvey et al., 2015).

Ren et al. (2020) present *neural IL* following the principles of Kirby (2001), where neural agents play a referential game and evolve a communication protocol through IL. They use topographic similarity (Brighton & Kirby, 2006) to quantify compositionality, and find that high topographic similarity improves the learning speed of neural agents, allows the listener to recognize more objects using less data, and increases validation performance. However, these experiments are limited to domains with extremely simple object and message structure.

Several ablation studies (Li & Bowling, 2019; Ren et al., 2020) have found that re-initializing the speaker and the listener between generations is necessary to reap the benefits of compositionality from IL. However, *seeded IL* (Lu et al., 2020) proposes to seed a new agent with the previous generation's parameters at the end of the learning phase of a new generation. Since self-play has not yet fine-tuned this initialization, it has not had the opportunity to develop a non-compositional language to fit the training data. The authors find that seeded IL helps counter language drift in a translation game, and hypothesize that IL maintains the compositionality of natural language.

## 3    METHOD

We are interested in solving the task of visual question-answering (VQA). Let $\mathcal{X}$ be the space of images about which our model will be required to answer questions. Next, let $\mathcal{Q}$ be the space of natural-language questions and $\mathcal{Y}$ the space of all possible answers to the questions. Additionally, we consider a space $\mathcal{Z}$ of programs, which represent computation graphs of operations that can be performed on an image in $\mathcal{X}$ to produce an output in $\mathcal{Y}$. We consider a question template $\boldsymbol{T}$ to be

a set of tuples $(\boldsymbol{q}, \boldsymbol{z})$, where $\boldsymbol{q} \in \mathcal{Q}$ and $\boldsymbol{z} \in \mathcal{Z}$. Each question template contains questions with the same structure but varying primitive values. For example, the questions "Is a triangle blue" and "Is a square red" belong to a template "Is a SHAPE COLOR." The program $\boldsymbol{z}$ corresponding to the question $\boldsymbol{q}$ in a template defines a computation graph of operations that would produce the correct answer in $\mathcal{Y}$ to $\boldsymbol{q}$ for any input image $\boldsymbol{x} \in \mathcal{X}$. Finally, let $\mathcal{T}$ be a finite set of question templates.

The dataset for training our model and evaluating VQA performance constitutes tuples of the form $(\boldsymbol{q}, \boldsymbol{z}, \boldsymbol{x}, y)$. First, a template $\boldsymbol{T} \in \mathcal{T}$ is sampled and a tuple $(\boldsymbol{q}, \boldsymbol{z})$ is sampled from $\boldsymbol{T}$. Then, an image $\boldsymbol{x} \in \mathcal{X}$ is sampled and the answer $y$ is produced by passing $\boldsymbol{x}$ through the program $\boldsymbol{z}$. These collected variables $(\boldsymbol{q}, \boldsymbol{z}, \boldsymbol{x}, y)$ form a single example in the task dataset. To evaluate our model's performance on unseen question templates, we define $\mathcal{T}_{train} \subset \mathcal{T}$ to be a subset of training templates and $\mathcal{T}_{test} = \mathcal{T} - \mathcal{T}_{train}$ to be the subset of test templates. The training dataset $\mathcal{D}$ is prepared from templates in $\mathcal{T}_{train}$ and the out-of-distribution test dataset $\mathcal{D}_{test}$ from templates in $\mathcal{T}_{test}$. We allow a program $\boldsymbol{z}$ to be *absent* in $\mathcal{D}$, in which case it is not used for auxiliary supervision during training.

Our goal of systematic generalization is to learn a model $p(\boldsymbol{Y} \mid \boldsymbol{X}, \boldsymbol{Q})$ that performs well on the dataset $\mathcal{D}_{test}$ created using unseen question templates, where $\boldsymbol{Y}$, $\boldsymbol{X}$, and $\boldsymbol{Q}$ are random variables taking values in $\mathcal{Y}$, $\mathcal{X}$, and $\mathcal{Q}$. We define our model to be a composition of a program generator $PG_\theta(\boldsymbol{Z} \mid \boldsymbol{Q})$ and an execution engine $EE_\phi(\boldsymbol{Y} \mid \boldsymbol{X}, \boldsymbol{Z})$, parameterized by $\theta$ and $\phi$ respectively.

### 3.1 Model architecture

Our overall model architecture is illustrated in Figure 1. We use the attention-based sequence-to-sequence model from Bahdanau et al. (2019a) as the program generator, which translates an input question $\boldsymbol{q}$ into a sampled program $\hat{\boldsymbol{z}}$. The execution engine assembles modules into a layout dictated by $\hat{\boldsymbol{z}}$ to instantiate an NMN that takes the input image $\boldsymbol{x}$ to predict an answer $\hat{y}$. In this work, we explore three execution engine architecture choices, including *Tensor-NMN*, which is the module architecture from Johnson et al. (2017b), and *Vector-NMN* from Bahdanau et al. (2019a). We also experiment with a novel hybrid of these architectures that performs better on SHAPES-SyGeT, which we dub *Tensor-FiLM-NMN*. Appendix B elaborates the details of our model architecture.

The goal of the Tensor-FiLM-NMN architecture is to combine the minimalist, FiLM-based definition of modules in Vector-NMN with the spatial representational power of Tensor-NMN. As such, the modules in Tensor-FiLM-NMN use FiLM to condition global operations on module-specific embeddings as in Vector-NMN, but the module inputs and outputs are tensor-valued feature maps like in Tensor-NMN. The following equations illustrate the working of a Tensor-FiLM-NMN module:

$$\tilde{\boldsymbol{h}}_1 = \boldsymbol{\gamma}_h \odot \boldsymbol{h}_1 \oplus \boldsymbol{\beta}_h, \quad \tilde{\boldsymbol{h}}_2 = \boldsymbol{\gamma}_h \odot \boldsymbol{h}_2 \oplus \boldsymbol{\beta}_h \tag{1}$$

$$\boldsymbol{e} = [(\boldsymbol{\gamma}_x \odot \boldsymbol{x} \oplus \boldsymbol{\beta}_x); \max(\tilde{\boldsymbol{h}}_1, \tilde{\boldsymbol{h}}_2); (\tilde{\boldsymbol{h}}_1 - \tilde{\boldsymbol{h}}_2)] \tag{2}$$

$$\boldsymbol{g} = \boldsymbol{W}_1 * \boldsymbol{e} \oplus \boldsymbol{b}_1 \tag{3}$$

$$\boldsymbol{y} = \text{ReLU}(\boldsymbol{W}_2 * (\boldsymbol{\gamma}_g \odot \text{ReLU}([\boldsymbol{g}; \text{cumsum\_left}(\boldsymbol{g}); \text{cumsum\_down}(\boldsymbol{g})]) \oplus \boldsymbol{\beta}_g) \oplus \boldsymbol{b}_2 + \boldsymbol{e}). \tag{4}$$

Here, $\boldsymbol{x}$ is the input image, $\boldsymbol{h}_1$ and $\boldsymbol{h}_2$ are the module inputs, and $\boldsymbol{y}$ is the module output. $\boldsymbol{\gamma}_\bullet$ and $\boldsymbol{\beta}_\bullet$ are FiLM parameters computed using different 2-layer MLPs per layer on the module-specific embeddings. The weights $\boldsymbol{W}_\bullet$ and biases $\boldsymbol{b}_\bullet$ are shared across all modules. We further strengthen our model's ability for spatial reasoning by using cumulative sums of an intermediate representation across locations in left-to-right and top-to-bottom directions. This allows our model, for example, to select regions to the 'left of' or 'below' objects through appropriate scaling and thresholding.

### 3.2 Iterated learning for NMNs

We can view the program generator and the execution engine as communicating agents in a cooperative VQA game where the programs passed between agents are messages drawn from an emergent language. Introducing more compositional structure to this language, such as reusing low-level concepts using the same tokens and high-level concepts using the same sequence of tokens, helps address the combinatorial complexity of the question space, allowing agents to perform better on new question templates containing previously-unseen combinations of known concepts.

We use IL to encourage the emergence of structure in the generated programs. We iteratively spawn new program generator and execution engine agents, train them on the VQA task, and transfer their

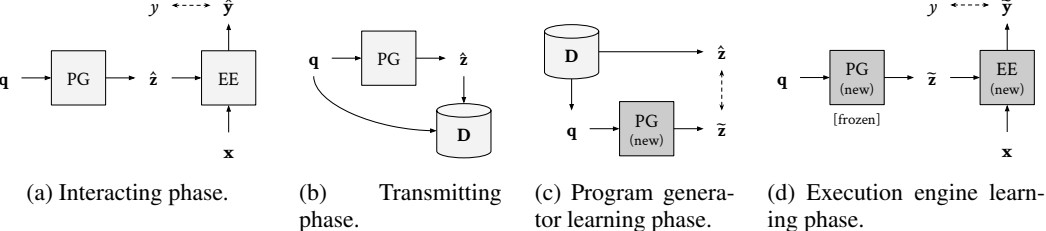

(a) Interacting phase.  (b) Transmitting phase.  (c) Program generator learning phase.  (d) Execution engine learning phase.

Figure 2: Phases of IL for emergent module layouts. Solid arrows indicate forward pass through the model, and dashed lines indicate the cross-entropy loss between predictions and targets. After proceeding through phases (a)-(d), the new program generator and execution engine begin an interacting phase (a) of a new generation.

knowledge to the next generation of agents. Limiting this transmission of knowledge between generations imposes a *learning bottleneck*, where only the easy-to-learn linguistic concepts survive. Since the central hypothesis of neural IL is that structure is easier for neural networks to learn than idiomatic non-compositional rules (Li & Bowling, 2019; Ren et al., 2020), our IL algorithm pushes the language of the programs to be more compositional. The combination of learning a compositional structure and performing well on the VQA training task engenders systematic generalization.

We follow Ren et al. (2020) in dividing our method into three stages, an *interacting* phase, a *transmitting* phase, and a *learning* phase, which are cycled through iteratively until the end of training. Figure 2 illustrates the phases of our method for training module networks with compositional emergent layouts and Appendix C presents a formal algorithm for the same.

**Interacting phase.** The program generator and the execution engine must work together. Without programs that consistently use the same tokens to mean the same operations, the execution engine cannot assign the correct semantics to modules. Simultaneously, the program generator depends on the language grounding provided by the execution engine in the form of reinforcement signal for beneficial layouts. To make the problem more tractable, it is common to provide the model a small set of ground-truth programs (Johnson et al., 2017b; Vedantam et al., 2019). We find that providing program supervision with a small number of ground-truth programs throughout the interacting phase helps in generalization. This can be seen as a form of supervised self-play (Lowe et al., 2020), and has the effect of *simulating* an inductive bias towards the desired program language.

The agents are jointly trained for $T_i$ steps to minimize answer prediction error on data sampled from the training dataset $\mathcal{D}$, with predictions given by $\hat{\boldsymbol{z}} \sim PG_\theta(\boldsymbol{q})$; $\hat{\boldsymbol{y}} = EE_\phi(\boldsymbol{x}, \hat{\boldsymbol{z}})$. We train $EE_\phi$ by minimizing the cross-entropy loss $L$ between the true answer label $y$ and the predicted distribution $\hat{\boldsymbol{y}}$. Additionally, we estimate the gradients for the parameters $\theta$ using REINFORCE:

$$\nabla_\theta L_{PG}(\theta) = \mathbb{E}_{\hat{\boldsymbol{z}} \sim p_\theta(\cdot | \boldsymbol{q})} \left[ \text{clip}(-L, -5, 5) \nabla_\theta \log p_\theta(\hat{\boldsymbol{z}} \mid \boldsymbol{q}) \right]. \tag{5}$$

Here, $p_\theta$ is the distribution over programs returned by $PG_\theta(\boldsymbol{q})$. We use the negative cross-entropy loss $-L$ clipped between $-5$ and $5$ as the reward for the program generator. When a ground-truth program $\boldsymbol{z}$ is available, we additionally minimize a weighted cross-entropy loss between $\boldsymbol{z}$ and the generated program $\hat{\boldsymbol{z}}$ using teacher forcing. In practice, the model operates on minibatches of data, and a subset of every minibatch is subsampled from the examples with ground-truth programs.

**Transmitting phase.** During the transmitting phase, a new dataset $\boldsymbol{D}$ with $T_t$ samples is prepared for use during the learning phase. Questions $\boldsymbol{q}$, as well as ground-truth programs $\boldsymbol{z}$ if available, are sampled from $\mathcal{D}$. For data examples without ground-truth programs, a program $\hat{\boldsymbol{z}}$ is sampled from the execution engine using $\boldsymbol{q}$. Finally, question-program pairs $(\boldsymbol{q}, \boldsymbol{z})$, if $\boldsymbol{z}$ is available, or $(\boldsymbol{q}, \hat{\boldsymbol{z}})$ are added to $\boldsymbol{D}$. By transmitting the ground-truth programs when available, we continue simulating the inductive bias towards desirable programs during the learning phase.

**Learning phase.** Program generators and execution engines are initialized in the learning phase, forming a new generation of agents to play the VQA game. The new program generator then acquires the previous agents' language from the transmitted data $\boldsymbol{D}$. However, it does so imperfectly due to a learning bottleneck that exerts pressure towards compositionality (Ren et al., 2020).

We implement this learning bottleneck primarily by limiting the number of gradient updates in the learning phase, effectively performing *early stopping*. A smaller number of program generator gradient updates $T_p$ leads to an underfit program generator that could waste computation during the interacting phase to re-learn global linguistic rules. On the other hand, setting $T_p$ to a high value can lead to overfitting and learning of the previous generation's idiosyncrasies, losing the benefits of the learning bottleneck. In addition to limiting the number of training iterations, we optionally further regularize the model by applying *spectral normalization* (Miyato et al., 2018) on the program generator's decoder LSTM parameters.

We train the new program generator $PG_{\tilde{\theta}}$ by minimizing the cross-entropy loss between model samples $\tilde{z} \sim PG_{\tilde{\theta}}(q)$ and transmitted programs $\hat{z}$ corresponding to $q$ in $D$ for $T_p$ gradient steps. Then, the new execution engine $EE_{\tilde{\phi}}$ is trained with programs from the new program generator $PG_{\tilde{\theta}}$, using the entire dataset $\mathcal{D}$ for $\tilde{T}_e$ steps. The forward pass in the execution engine learning phase is similar to that in the interacting phase, but the backward pass only updates the execution engine parameters $\tilde{\phi}$. Adapting the new execution engine to the new program generator ensures stable training in the upcoming interacting phase.

## 4 SYSTEMATIC GENERALIZATION TEST FOR SHAPES

The SHAPES dataset (Andreas et al., 2016) is a popular yet simple VQA dataset. The lower image and question complexities of SHAPES relative to CLEVR make it an attractive choice for experimentation, as training can be faster and require fewer resources. Each of the $15616$ data points in SHAPES contains a unique image. However, there are only $244$ unique questions. Although the validation and test splits of SHAPES contain unique questions not present in any of the other splits, the questions are of the same form as the ones in the training split.

To the best of our knowledge, there is no published systematic generalization evaluation setup for SHAPES. Thus, we present *SHAPES-SyGeT*, a **SHAPES Sy**stematic **Ge**neralization **T**est[1]. To understand the distribution of question types in SHAPES, we categorized questions into $12$ standard templates, out of which $7$ are used for training, and $5$ to test systematic generalization.

To evaluate the in-distribution and out-of-distribution generalization performance of our models, we prepare the SHAPES-SyGeT training set with only a subset of the questions under each train template and use the rest as an in-distribution validation set (*Val-IID*). Questions belonging to the evaluation templates are used as an out-of-distribution validation set (*Val-OOD*). Please see Appendix D for further details about the question templates and the dataset splits.

## 5 EXPERIMENTS

In this section, we present results on SHAPES-SyGeT, CLEVR, and CLOSURE. For our preliminary experiments on GQA (Hudson & Manning, 2019), please see Appendix G. The SHAPES-SyGeT experiments are run with 3 different seeds and the CLEVR experiments use 8 seeds. We report the mean and standard deviation of metrics over these trials. CLEVR experiments use representations of images from a pre-trained ResNet-101 (He et al., 2016) as input, whereas the SHAPES-SyGeT runs use standardized raw images. Please see Appendix A for the hyperparameters we use.

For our NMN baselines without IL, we still utilize the multi-task objective of using REINFORCE to backpropagate gradients to the program generator and using a cross-entropy loss to provide program supervision when available. We find this method of training NMNs to generalize better than pre-training on the available programs and then interacting without supervision as is done in Johnson et al. (2017b). One can also view our baselines as executing one long interacting phase without any other phases or resetting of the model.

### 5.1 SHAPES-SYGET

Due to its lightweight nature, we use SHAPES-SyGet for a series of experiments designed to illuminate the essential components of our method. For all SHAPES-SyGeT experiments, we record the

---

[1]SHAPES-SyGeT can be downloaded from: `https://github.com/ankitkv/SHAPES-SyGeT`.

Table 1: SHAPES-SyGeT accuracies. NMNs are trained with 20 or 135 ground-truth programs; FiLM (Perez et al., 2018) and MAC (Hudson & Manning, 2018) do not use program supervision.

| Model | Val-IID | | Val-OOD | |
|---|---|---|---|---|
| FiLM | $0.720 \pm 0.01$ | | $\mathbf{0.609 \pm 0.01}$ | |
| MAC | $\mathbf{0.730 \pm 0.01}$ | | $0.605 \pm 0.01$ | |
| | **#GT 20** | **#GT 135** | **#GT 20** | **#GT 135** |
| Tensor-NMN | $0.645 \pm 0.01$ | $0.700 \pm 0.01$ | $0.616 \pm 0.01$ | $0.641 \pm 0.03$ |
| Tensor-NMN+IL | $0.756 \pm 0.07$ | $0.763 \pm 0.04$ | $0.648 \pm 0.02$ | $0.661 \pm 0.02$ |
| Tensor-FiLM-NMN | $0.649 \pm 0.02$ | $0.851 \pm 0.01$ | $0.605 \pm 0.01$ | $0.692 \pm 0.06$ |
| Tensor-FiLM-NMN+IL | $\mathbf{0.954 \pm 0.07}$ | $\mathbf{1.000 \pm 0.00}$ | $\mathbf{0.858 \pm 0.15}$ | $\mathbf{0.971 \pm 0.02}$ |

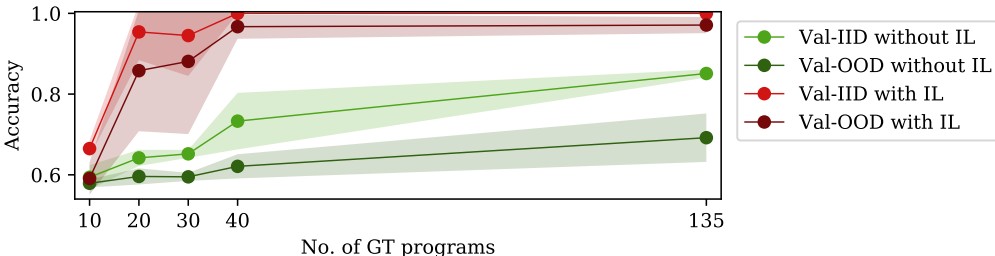

Figure 3: SHAPES-SyGeT answer accuracies of models with varying number of ground-truth programs, with and without IL. All models use a Tensor-FiLM-NMN execution engine.

in-distribution Val-IID and the out-of-distribution Val-OOD accuracies. However, model selection and early stopping are done based on Val-IID. For our IL experiments, we use spectral normalization of the program generator during the learning phase, and we study its utility in Appendix E. We do not report results on Vector-NMN, which performs very poorly. We believe that this is because SHAPES-SyGeT requires intermediate module outputs to contain detailed spatial information, which Vector-NMN has an inductive bias against thanks to its module outputs being vectors.

We note that the reported results on SHAPES in the literature make use of NMN architectures with specially designed modules for each operation (Andreas et al., 2016; Hu et al., 2017; Vedantam et al., 2019). In contrast, we use generic modules in order to study compositional layout emergence with minimal semantic priors, making our results hard to compare directly to prior work using SHAPES.

**In-distribution and out-of-distribution accuracies.** Table 1 illustrates the difference in the performance of various configurations of our models. All the reported models achieve perfect accuracy on the training data but generalize differently to Val-IID and Val-OOD. We find that Tensor-FiLM-NMN generalizes better than Tensor-NMN trained with supervision of both 20 and 135 ground-truth programs. Moreover, we find that IL improves generalization across all configurations. We further evaluate the relationship between program supervision and performance in Figure 3 and find that IL improves performance at all levels of supervision, effectively improving data efficiency.

**Learning bottleneck.** Although all models achieve perfect training accuracy in under 5000 steps, we notice that only IL leads to gradually increasing systematic generalization as training progresses. Figure 4 shows the effect of the learning bottleneck of IL, comparing two module architectures with and without IL. In the presence of some ground-truth program supervision that the emergent language should abide by, there is a stricter notion of correct programs[2]. Thus, in calculating the program accuracy, we consider a program to be correct if it matches the ground-truth program exactly and incorrect otherwise. We find that the program accuracy increases steadily through the

---

[2]It is unlikely that a diverse set of programs provided for supervision could fit into a compositional structure significantly different from the ground-truth. Furthermore, small errors in the serialized programs could result in drastic differences in the parsed computation graph structure and semantics.

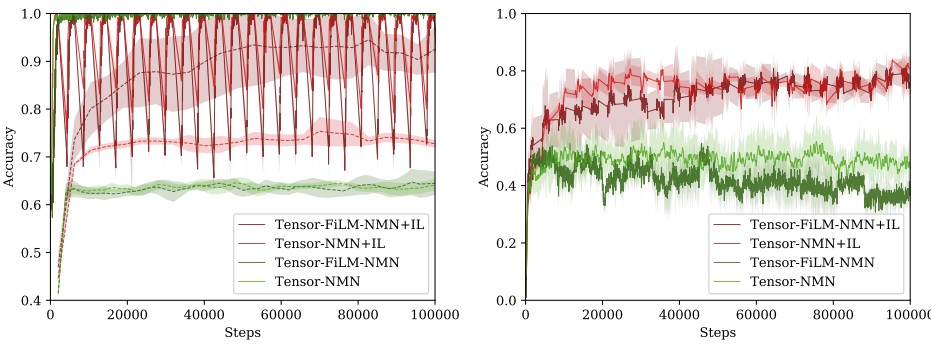

(a) Task accuracy. Solid lines are training and dashed lines are Val-IID.

(b) Program accuracy.

Figure 4: Learning curves of models with and without IL on SHAPES-SyGeT, using 20 ground-truth programs. The IL curves use a global gradient step counter across all phases. The dips in the IL training curves indicate the beginning of new generations.

influence of the learning bottleneck in the case of IL as training progresses, indicating increasing structure in the language of the programs.

**Ablation tests.** We use SHAPES-SyGeT to perform a series of ablations tests and report the full results for these experiments in Appendix E. We first examine the importance of reinitializing the execution engine. We consider two alternatives to reinitializing it from scratch at every generation: seeded IL (Lu et al., 2020), which uses the parameters at the end of the previous execution engine learning phase for re-initialization, and simply retaining the execution engine without any re-initialization. We find that both of these methods harm generalization performance compared to the standard setting.

Next, we study the effect of spectral normalization applied to the program generator decoder. Without IL, spectral normalization has a marginal effect on generalization, indicating it alone is not sufficient to achieve the generalization improvements we observe. However, it improves the Val-IID and Val-OOD accuracies substantially with IL in the standard execution engine setting. Finally, we observe that retaining the previous program generator in a new generation without re-training greatly harms the performance of our best model, indicating that the learning phase is crucial.

Curiously, we note that the effects of spectral normalization and not resetting the program generator are small or reversed when the execution engine is not re-initialized from scratch. This can be explained by an empirical observation that a partially trained execution engine finds it easier to overfit to the input images when the programs are noisy in the case of a small dataset like SHAPES-SyGeT, instead of waiting for the program generator to catch up to the existing module semantics.

## 5.2 CLEVR AND CLOSURE

CLEVR is significantly larger and more complex than SHAPES-SyGeT. It takes an execution engine over a day to reach $95\%$ task accuracy on the CLEVR validation set on a Nvidia RTX-8000 GPU, even when trained with ground-truth programs without a program generator. Re-training the execution engine from scratch for every generation of IL is thus computationally infeasible. Between using seeded IL and not resetting the execution engine at all, we find not resetting the execution engine generalizes better. CLEVR contains 699989 training questions, and we provide 100 ground-truth programs for program supervision, one-tenth of that used by Vedantam et al. (2019). We find that Tensor-FiLM-NMN does not improve over Vector-NMN for CLEVR. Thus, we report results only on Tensor-NMN and Vector-NMN for conciseness.

Figure 5 illustrates the training dynamics with and without IL on CLEVR, and Table 2 reports model performance on the CLEVR validation set and the out-of-distribution CLOSURE categories. The CLEVR validation curves through training are presented in Appendix F. Similar to Bahdanau et al. (2019a), we find that Vector-NMN generalizes better than Tensor-NMN on CLEVR without IL.

Table 2: Task accuracy on the CLEVR validation set and each CLOSURE category for models trained with and without IL, using 100 ground-truth programs.

| Evaluation set | Tensor-NMN | | Vector-NMN | |
|---|---|---|---|---|
| | Without IL | With IL | Without IL | With IL |
| CLEVR-Val | $0.912 \pm 0.07$ | $\mathbf{0.964 \pm 0.01}$ | $0.960 \pm 0.01$ | $\mathbf{0.964 \pm 0.00}$ |
| and_mat_spa | $\mathbf{0.278 \pm 0.17}$ | $0.264 \pm 0.16$ | $\mathbf{0.400 \pm 0.13}$ | $0.335 \pm 0.18$ |
| or_mat | $0.327 \pm 0.11$ | $\mathbf{0.481 \pm 0.24}$ | $0.367 \pm 0.11$ | $\mathbf{0.563 \pm 0.23}$ |
| or_mat_spa | $0.286 \pm 0.13$ | $\mathbf{0.405 \pm 0.22}$ | $0.330 \pm 0.11$ | $\mathbf{0.444 \pm 0.24}$ |
| compare_mat | $0.793 \pm 0.11$ | $\mathbf{0.851 \pm 0.17}$ | $0.660 \pm 0.16$ | $\mathbf{0.873 \pm 0.12}$ |
| compare_mat_spa | $0.746 \pm 0.13$ | $\mathbf{0.853 \pm 0.15}$ | $0.677 \pm 0.14$ | $\mathbf{0.871 \pm 0.12}$ |
| embed_spa_mat | $0.824 \pm 0.07$ | $\mathbf{0.947 \pm 0.03}$ | $0.863 \pm 0.07$ | $\mathbf{0.900 \pm 0.08}$ |
| embed_mat_spa | $0.739 \pm 0.14$ | $\mathbf{0.941 \pm 0.02}$ | $0.894 \pm 0.03$ | $\mathbf{0.936 \pm 0.03}$ |

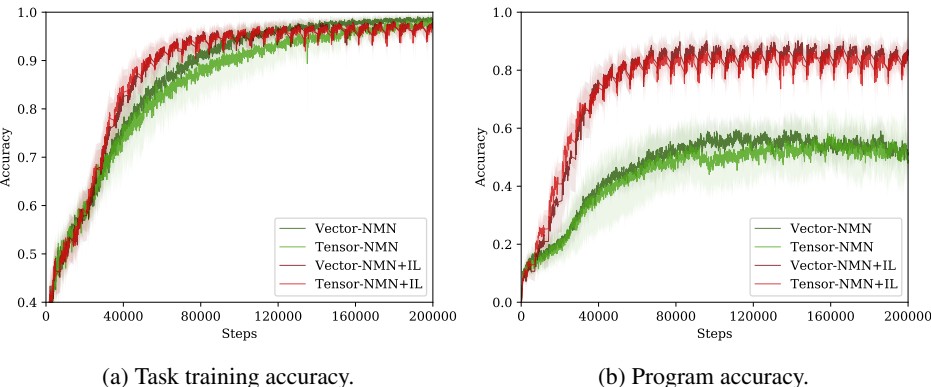

(a) Task training accuracy.      (b) Program accuracy.

Figure 5: Learning curves of models trained with and without IL on CLEVR using 100 ground-truth programs. The IL curves use a global gradient step counter across all phases. The dips in the IL training curves indicate the beginning of new generations.

However, Tensor-NMN systematically generalizes substantially better with IL. Across both models, using IL improves generalization on CLOSURE, greatly increases program accuracy, and achieves validation performance on CLEVR close to ProbNMN (Vedantam et al., 2019) despite using far fewer programs for supervision.

## 6 CONCLUSION

We establish IL as a practical tool for the emergence of structure in machine learning by demonstrating its utility in challenging VQA tasks using NMNs. Our work shows that IL leads to improved performance with less supervision and facilitates systematic generalization. As the study of IL in neural agents is relatively young and has largely been explored in the narrow domain of language emergence in referential games, our setting presents several important new challenges for IL methods. Unlike in simple games (Ren et al., 2020), the emergence of compositional language without any supervision is thus far infeasible in VQA due to the difficult joint optimization of the program generator and the execution engine. However, by exploring learning phase regularization, suitable model architectures, and the learning dynamics of IL, we are able to dramatically reduce the amount of ground-truth supervision necessary; the surprising success of spectral normalization in particular should motivate further research into the role of regularization during the learning phase. We hope that this progress will spur future work to improve IL methods and their theoretical understanding, as well as extend them to other challenging tasks in machine learning.

ACKNOWLEDGMENTS

We acknowledge the financial support of Samsung, Microsoft Research, and CIFAR.

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

Table 3: IL hyperparameters used in our experiments.

| Hyperparameter | Value |
|---|---|
| Optimizer | Adam (Kingma & Ba, 2015) |
| PG learning rate | 0.0001 |
| EE learning rate | 0.0001 for CLEVR, 0.0005 for SHAPES-SyGeT Tensor-NMN, 0.001 for Tensor-FiLM-NMN |
| Batch size | 128 |
| GT programs in batch | 4 |
| PG REINFORCE weight | 10.0 |
| PG GT cross-entropy weight | 1.0 |
| PG spectral normalization | On for SHAPES-SyGeT, off for CLEVR |
| Interacting phase length $T_i$ | 2000 for SHAPES-SyGeT, 5000 for CLEVR |
| Transmitting dataset size $T_t$ | $2000 \times$ batch size $= 256000$ |
| PG learning phase length $T_p$ | 2000 |
| EE learning phase length $T_e$ | For SHAPES-SyGeT, 250 when re-initializing EE from scratch, 200 for seeded IL; For both SHAPES-SyGeT and CLEVR, 50 when not resetting EE |

## A  HYPERPARAMETERS

Table 3 details the hyperparameters used in our experiments in Section 5 for both SHAPES-SyGeT and CLEVR.

## B  MODEL ARCHITECTURE DETAILS

### B.1  PROGRAM GENERATOR

The program generator is a seq2seq model that translates input questions to programs. We use the Seq2SeqAtt model of Bahdanau et al. (2019a), where unlike Johnson et al. (2017b), the decoder uses an attention mechanism (Bahdanau et al., 2015) over the encoder hidden states. We found this model to generalize better on held-out in-distribution questions in preliminary experiments. Ideally, the program generator's target must be tree-structured since the output program represents a computation graph. Since a seq2seq model cannot directly model graphs, we instead model a serialized representation of the programs. Like Johnson et al. (2017b), we chose to use the Polish (or prefix) notation to serialize the syntax tree.

Considering the program generator parameterized by $\theta$, the probability distribution over layouts $\hat{z}$ is given by

$$p_\theta(\hat{z} \mid \boldsymbol{q}) = \prod_{i=1}^{t} p_\theta(\hat{z}_i \mid \hat{z}_{1:i-1}, \boldsymbol{q}). \tag{6}$$

During training, we sample programs from this distribution but take the $\arg\max$ at each timestep when evaluating our model. We use an LSTM (Hochreiter & Schmidhuber, 1997) trained using teacher forcing to represent the distribution $p_\theta(\hat{z}_i \mid \hat{z}_{1:i-1}, \boldsymbol{q})$ at every timestep $i$. To condition on $\boldsymbol{q}$ for each decoder step, we perform attention on the hidden states of a bidirectional LSTM encoder over the question $\boldsymbol{q}$.

### B.2  EXECUTION ENGINE

To run the program $\hat{z}$ produced by the program generator, the execution engine assembles neural modules according to $\hat{z}$ into a neural network that takes $\boldsymbol{x}$ as input to produce an answer $\hat{y}$. We consider this execution engine $EE_\phi(\boldsymbol{x}, \hat{z})$ to be parameterized by $\phi$. To illustrate a forward pass through the module network, consider the program 'and color[green] scene transform[left_of] shape[square] scene' from Figure 1. 'scene' is a special module which represents the input image features from a CNN feature extractor. According to the

annotators, the 'color[green]' module should filter green shapes from the scene, and the 'shape[square]' module should filter squares. 'transform[left_of]' should select regions to the left of the filtered squares. In Figure 1, the final 'and' module should then compute the intersection of green shapes and the region to the left of squares. A classifier takes this output from the top-level module to produce a distribution $\hat{y}$ over answers. We consider the final answer $\hat{y}$ to be the $\arg\max$ of $\hat{y}$.

Each module corresponds to one token of the program and can be reused in different configurations. Like Johnson et al. (2017b), our modules are not hand-designed to represent certain functions but have a similar architecture for all operators. Other than the 'scene' module which has arity 0, all other modules are implemented as a CNN on the module inputs to produce a module output. In this work, we explore three module architecture choices to make up the execution engine: *Tensor-NMN*, *Vector-NMN*, and *Tensor-FiLM-NMN*. Here, the Tensor-NMN module architecture is the module network proposed by Johnson et al. (2017b), Vector-NMN is the architecture from Bahdanau et al. (2019a), and Tensor-FiLM-NMN is described in Section 3.1.

## C    ITERATED LEARNING ALGORITHM

We present the full IL algorithm described in Section 3.2 in Algorithm 1.

## D    SHAPES-SYGET TEMPLATES AND SPLITS

The images in SHAPES are arranged in a $3 \times 3$ grid, where each grid cell can contain a triangle, square, or circle that is either red, green, or blue, or the cell can be empty. Table 4a shows the standard splits for training and evaluating SHAPES used in the literature (Andreas et al., 2016; Hu et al., 2017; Vedantam et al., 2019). We categorize SHAPES questions into 12 standard templates, listed in Table 5. We then derive new splits for the SHAPES data, such that each split has the same primitives, but different ways in which the primitives are combined. The final split of the SHAPES-SyGeT dataset is presented in Table 4b.

## E    ABLATION EXPERIMENTS

We perform ablation experiments on SHAPES-SyGeT rather than CLEVR/CLOSURE as the computational requirements of experiments on this dataset are an order of magnitude lighter. Results are presented in Table 6. We focus primarily on Tensor-FiLM-NMN, which we find benefits far more from IL than Tensor-NMN. In all cases, we use 20 ground-truth programs for supervision, a choice which we find to clearly reflect the differences between the various settings considered. We describe the important observations from our ablation study in Section 5.1.

## F    CLEVR VALIDATION CURVES

The validation curves for CLEVR, illustrated in Figure 6, are similar to the training curves in Figure 5. Although CLEVR training exhibits a lower amount of overfitting compared to SHAPES-SyGeT, we observe higher systematic generalization performance on CLOSURE with models trained using IL.

## G    PRELIMINARY EXPERIMENTS WITH GQA

To study if IL can offer benefits for larger datasets without synthetic images and questions, we choose to evaluate our IL method on the GQA dataset (Hudson & Manning, 2019). As a preliminary setup, we remain as close as possible to the architectural setup described in Section 3. We only use question text, spatial image features from a pre-trained ResNet-101 (He et al., 2016), and programs, all without object annotations. We serialize the programs according to the Polish notation as we do in the case of SHAPES-SyGeT and CLEVR, which can result in duplicated token sequences as GQA programs do not always form strict directed acyclic graphs. Although these limitations allow us to easily apply the presented method to GQA, they also prevent us from getting competitive results on

---

**Algorithm 1:** Iterated learning for compositional NMN layout emergence.

---

**Data:** $N_{epochs}$: number of epochs, $(T_i, T_t, T_p, T_e)$: length of each phase, $\mathcal{D}$: training dataset.

1 Initialize $PG$ parameters $\theta_1$ and $EE$ parameters $\phi_1$;

2 **for** $n = 1$ **to** $N_{epochs}$ **do**

    `/* Interacting phase                                                */`

3     **for** $i = 1$ **to** $T_i$ **do**

4         Sample new tuple $(\boldsymbol{q}, \boldsymbol{z}, \boldsymbol{x}, y)$ from $\mathcal{D}$;

5         $\hat{\boldsymbol{z}} \sim PG_{\theta_n}(\boldsymbol{q})$;

6         $L \leftarrow$ cross-entropy-loss$(y, EE_{\phi_n}(\boldsymbol{x}, \hat{\boldsymbol{z}}))$;

7         **if** $\boldsymbol{z}$ *is available* **then** $L \leftarrow L +$ cross-entropy-loss$(\boldsymbol{z}, \hat{\boldsymbol{z}})$ ;

8         Update $\theta_n$ and $\phi_n$ to minimize $L$;

9     **end**

    `/* Transmitting phase                                               */`

10     $\boldsymbol{D}' = \varnothing$;

11     **for** $i = 1$ **to** $T_t$ **do**

12         sample new tuple $(\boldsymbol{q}, \boldsymbol{z}, \boldsymbol{x}, y)$ from $\mathcal{D}$;

13         **if** $\boldsymbol{z}$ *is available* **then** $\hat{\boldsymbol{z}} \leftarrow \boldsymbol{z}$ ;

14         **else** $\hat{\boldsymbol{z}} \sim PG_{\theta_n}(\boldsymbol{q})$ ;

15         Add $(\boldsymbol{q}, \hat{\boldsymbol{z}})$ to $\boldsymbol{D}'$;

16     **end**

    `/* Program generator learning phase                                 */`

17     Initialize $PG$ parameters $\theta_{n+1}$;

18     **for** $i = 1$ **to** $T_p$ **do**

19         Sample new tuple $(\boldsymbol{q}, \hat{\boldsymbol{z}})$ from $\boldsymbol{D}'$;

20         $\tilde{\boldsymbol{z}} \sim PG_{\theta_{n+1}}(\boldsymbol{q})$;

21         $L \leftarrow$ cross-entropy-loss$(\hat{\boldsymbol{z}}, \tilde{\boldsymbol{z}})$;

22         Update $\theta_{n+1}$ to minimize $L$ with spectral normalization;

23     **end**

    `/* Execution engine learning phase                                  */`

24     Initialize $EE$ parameters $\phi_{n+1}$;

25     **for** $i = 1$ **to** $T_e$ **do**

26         Sample new tuple $(\boldsymbol{q}, \boldsymbol{z}, \boldsymbol{x}, y)$ from $\mathcal{D}$;

27         $\tilde{\boldsymbol{z}} \sim PG_{\theta_{n+1}}(\boldsymbol{q})$;

28         $L \leftarrow$ cross-entropy-loss$(y, EE_{\phi_{n+1}}(\boldsymbol{x}, \tilde{\boldsymbol{z}}))$;

29         Update $\phi_{n+1}$ to minimize $L$;

30     **end**

31 **end**

---

| Split | Total questions | Unique questions |
|---|---|---|
| Train | 13568 | 212 |
| Val | 1024 | 16 |
| Test | 1024 | 16 |

(a) SHAPES dataset.

| Split | Total questions | Unique questions |
|---|---|---|
| Train | 7560 | 135 |
| Val-IID | 1080 | 135 |
| Val-OOD | 6976 | 109 |

(b) SHAPES-SyGeT dataset.

Table 4: Split of questions in the SHAPES and SHAPES-SyGeT datasets.

| Template | Total questions | Unique questions |
|---|---|---|
| **Train templates** | 8640 | 135 |
| 1.  is a `COLOR` shape `RELATIVE(1)` a `COLOR` shape | 2304 | 36 |
| 2.  is a `SHAPE` `RELATIVE(1)` a `COLOR` shape | 2304 | 36 |
| 3.  is a `SHAPE` `RELATIVE(2)` a `COLOR` shape | 960 | 15 |
| 4.  is a `SHAPE` `RELATIVE(2)` a `SHAPE` | 1344 | 21 |
| 5.  is a `COLOR` shape a `SHAPE` | 576 | 9 |
| 6.  is a `SHAPE` `COLOR` | 576 | 9 |
| 7.  is a `SHAPE` a `SHAPE` | 576 | 9 |
| **Evaluation templates** | 6976 | 109 |
| 8.  is a `COLOR` shape `RELATIVE(2)` a `COLOR` shape | 960 | 15 |
| 9.  is a `COLOR` shape `RELATIVE(1)` a `SHAPE` | 2304 | 36 |
| 10. is a `COLOR` shape `RELATIVE(2)` a `SHAPE` | 832 | 13 |
| 11. is a `SHAPE` `RELATIVE(1)` a `SHAPE` | 2304 | 36 |
| 12. is a `COLOR` shape `COLOR` | 576 | 9 |

Table 5: Overview and splits of the SHAPES-SyGeT templates. The placeholder `COLOR` can take values in 'red,' 'green,' and 'blue,' and `SHAPE` can be either 'circle,' 'triangle,' or 'square.' `RELATIVE(1)` is a placeholder for one of the positional prepositions 'above,' 'below,' 'left of,' or 'right of,' whereas `RELATIVE(2)` is a combination of two `RELATIVE(1)`s, such as 'above left of.'

the dataset. However, our goal is not to pursue state-of-the-art performance on GQA, but instead to study if we can still observe the advantage of IL in a limited ground-truth program setting for our NMN setup. Still, due to the significantly higher complexity of GQA, a few modifications and special considerations become necessary.

### G.1 CHANGES TO THE MODEL ARCHITECTURE

In both SHAPES-SyGeT and CLEVR, the programs are sequences of tokens where each token represents an operation, corresponding to one module of the assembled NMN. Each possible operation is carried out on the module input, the input image, or a combination of both, and has a fixed arity. However, the structure of programs in GQA has additional levels of granularity. Each timestep has an operation (*e.g.* `exist`, `query`), an optional sub-operation (*e.g.* `material`, `weather`), an integer arity, and a list of argument tuples. Each argument tuple in this list contains an argument (*e.g.* `horse`, `water`) as well as a Boolean value indicating if the argument is negated. We ignore additional details in the programs provided in the GQA dataset. As an illustration, the question "does the lady to the left of the player wear a skirt?" translates to the program "`verify.rel(skirt,wearing,o){1} relate(lady,to_the_left_of,s){1} select(player){0}`", containing three tokens with arities 1, 1, and 0 respectively. The operations and the corresponding sub-operations are separated by a period (`.`), and the parentheses contain the list of arguments, none of which are negated in this example.

Since every timestep of the program sequence is given by a tuple *(Op, Subop, Args, ArgNegs, Arity)*, we modify the program generator to produce this tuple instead of a single operation. For *Args* and

| PG: Seq2SeqAtt | EE: Tensor-NMN | Val-IID | Val-OOD |
|---|---|---|---|
| +FullSN | - | $0.646 \pm 0.03$ | $0.614 \pm 0.01$ |
| +NoSN | - | $0.645 \pm 0.01$ | $0.616 \pm 0.01$ |
| | EE: Tensor-FiLM-NMN | | |
| +FullSN | - | $0.642 \pm 0.02$ | $0.596 \pm 0.02$ |
| +NoSN | - | $0.649 \pm 0.02$ | $0.605 \pm 0.01$ |
| +IL | +IL | $0.954 \pm 0.07$ | $0.858 \pm 0.15$ |
| +IL+NoSN | +IL | $0.803 \pm 0.13$ | $0.630 \pm 0.07$ |
| +IL+NoRetrain | +IL | $0.834 \pm 0.13$ | $0.664 \pm 0.06$ |
| +IL | +IL+Seeded | $0.792 \pm 0.05$ | $0.626 \pm 0.01$ |
| +IL+NoSN | +IL+Seeded | $0.744 \pm 0.13$ | $0.647 \pm 0.12$ |
| +IL+NoRetrain | +IL+Seeded | $0.794 \pm 0.12$ | $0.662 \pm 0.11$ |
| +IL | +IL+NoReset | $0.624 \pm 0.02$ | $0.586 \pm 0.01$ |
| +IL+NoSN | +IL+NoReset | $0.629 \pm 0.01$ | $0.588 \pm 0.01$ |
| +IL+NoRetrain | +IL+NoReset | $0.682 \pm 0.05$ | $0.600 \pm 0.01$ |

Table 6: Ablation experiments for models on SHAPES-SyGeT trained with 20 ground-truth programs. 'FullSN': spectral normalization applied throughout training; "NoSN": no spectral normalization in any phase; 'NoRetrain': new program generator is taken from the previous generation without any re-training; 'Seeded': new execution engine is initialized to the state after the previous learning phase; 'NoReset': execution engine is not re-initialized at the start of a generation.

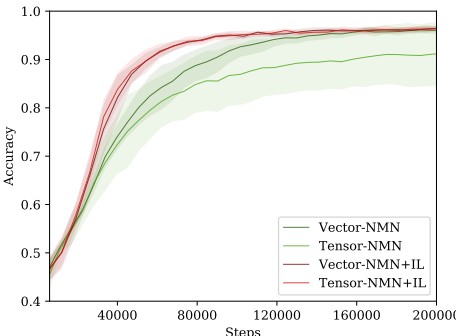

Figure 6: Validation curves of models trained with and without IL on CLEVR using 100 ground-truth programs.

*ArgNegs*, we consider a fixed argument list length of 3, padding with the argument <NULL> and no negation when the argument list is shorter than 3. This allows us to represent the distribution over the tokens at various levels of granularity for every timestep using a fixed number of logits.

Finally, it is no longer feasible to have dedicated modules for every possible unique program step, as the number of possible *(Op, Subop, Args, ArgNegs)*[3] combinations is very large. Thus, we restrict ourselves to using the Vector-NMN architecture, where instead of using a FiLM embedding for the operation the module represents, we concatenate embeddings for the *Op*, *Subop*, *Args*, and *ArgNegs* to generate timestep-specific FiLM embeddings.

## G.2 CHANGES TO THE METHOD

Unlike the question and program structures of SHAPES-SyGeT and CLEVR, the structures in GQA are significantly more diverse and include larger vocabularies as illustrated in Table 7. This makes training the program generator to a decent performance, even when trained on ground-truth programs directly, prohibitively slow. As a result, it becomes impractical to train the program generator

---

[3]*Arity* is consumed during the assembly of the NMN.

| Dataset | Token type | Vocabulary size |
|---|---|---|
| SHAPES-SyGeT | Question | 18 |
| | Program | 16 |
| CLEVR | Question | 93 |
| | Program | 44 |
| GQA | Question | 2939 |
| | Program *Op* | 16 |
| | Program *Subop* | 55 |
| | Program *Arg* | 2659 |

Table 7: Vocabulary sizes for SHAPES-SyGeT, CLEVR, and GQA token types after pre-processing.

| #GT programs | Model | Accuracy |
|---|---|---|
| - | `Vector-NMN` on GT programs | $0.556 \pm 0.002$ |
| 943000 | `Vector-NMN` | $0.486 \pm 0.002$ |
| 4000 | `Vector-NMN` | $0.455 \pm 0.006$ |
| 4000 | `Vector-NMN`+ResetEE | $0.469 \pm 0.007$ |
| 4000 | `Vector-NMN`+ResetEE+FinetuneEE | $0.470 \pm 0.007$ |
| 4000 | `Vector-NMN`+IL | $0.480 \pm 0.003$ |

Table 8: GQA validation accuracies. In Vector-NMN on GT programs, the execution engine always assembles modules according to the ground-truth programs, including at test time. For all the models except Vector-NMN+IL, the program generator is retained between generations. 'ResetEE': reset parameters except the FiLM embeddings; 'FinetuneEE': train the execution engine on a frozen program generator before joint training; 'IL': the full IL algorithm.

from scratch in every generation of IL. We thus need to explore strategies of resetting the program generator between generations while maintaining the benefits of the learning bottleneck of IL.

We hypothesize that due to the larger vocabulary sizes, learning good embeddings for question tokens and the various program tokens at all levels of granularity constitutes a large portion of the program generator training time. Following this intuition, we find it helpful to retain the input and output embeddings of the questions and the programs while resetting other parameters between generations. Experimentally, we verified that this method of resetting generalizes better than not resetting the program generator at all during IL. It also outperforms an alternate strategy of choosing new parameters based on an interpolation of a freshly initialized program generator and the final program generator from a generation.

Finally, despite a search over the Vector-NMN hyperparameters, we find that the optimal hyperparameters have a tendency to exhibit overfitting on the GQA images. The strategy of retaining embeddings and resetting the rest of the parameters also works well to combat this overfitting for the execution engine, where the embeddings we retain are the various FiLM embeddings. In Section G.3, we compare our IL method with baselines that only implement this partial reset of the execution engine as a regularizer.

## G.3 RESULTS

For our GQA experiments, we use the balanced train and validation splits for training and reporting results respectively. For each run, we report the mean and standard deviation across 3 trials. To study the effect of IL with limited ground-truth programs, we run our IL experiments with 4000 ground-truth programs, which constitutes only 0.4% of all available training programs.

Table 8 presents the results for different configurations of our models. A Vector-NMN execution engine that assembles modules using ground-truth programs during both training and validation provides an upper bound of performance over a learned program generator. When learning the

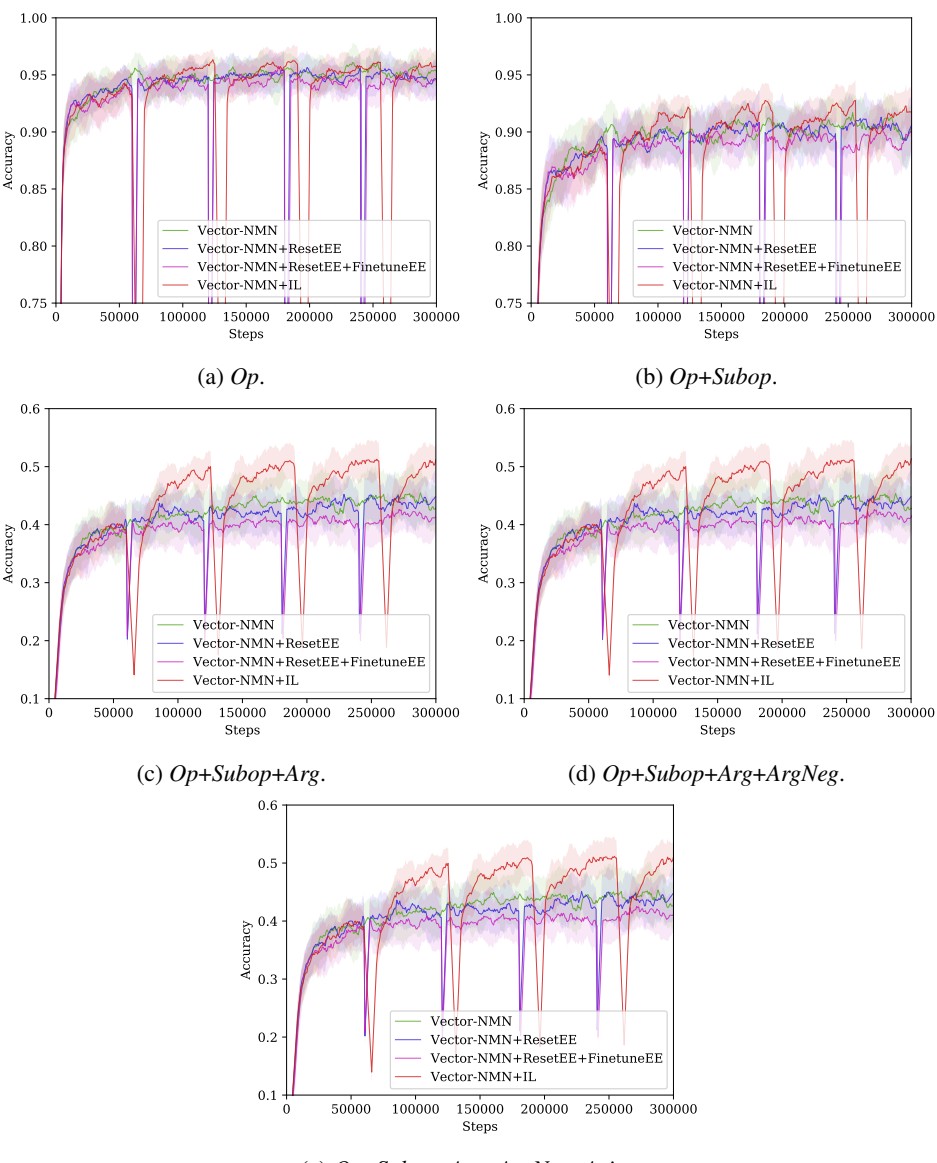

(a) *Op*.

(b) *Op+Subop*.

(c) *Op+Subop+Arg*.

(d) *Op+Subop+Arg+ArgNeg*.

(e) *Op+Subop+Arg+ArgNeg+Arity*.

Figure 7: Program accuracies at various levels of granularity of models trained on GQA using 4000 ground-truth programs. Here, (e) uses the strictest notion of program accuracy. The models with any form of IL use a global gradient step counter across all phases. The dips in the curves indicate the beginning of new generations.

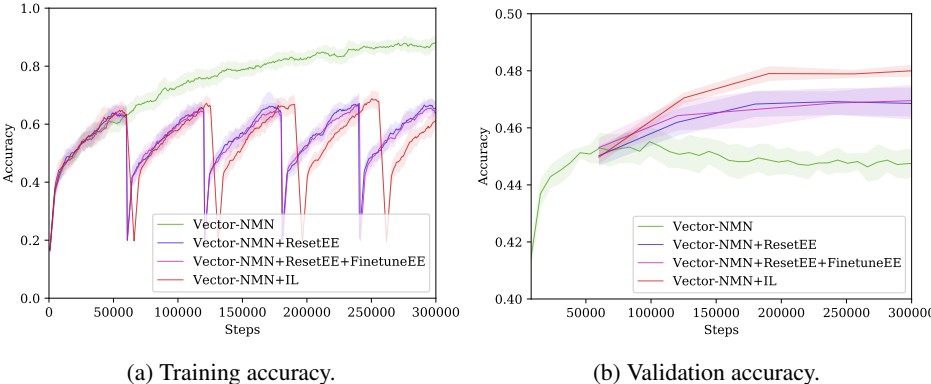

(a) Training accuracy.

(b) Validation accuracy.

Figure 8: Training and validation curves of models trained on GQA using 4000 ground-truth programs. The models with any form of IL use a global gradient step counter across all phases. The dips in the training curves indicate the beginning of new generations.

program generator, we find that using 4000 ground-truth programs with IL performs almost as well as using 943000 ground-truth programs without IL. Due to the tendency of the execution engine to overfit on the GQA images, we also compare our IL model with stronger baselines that perform regularization through iterated partial resetting of the execution engine. While we find this method to generalize better than standard training, we still find it necessary to employ the learning bottleneck through the learning phase of the program generator to achieve the best accuracy in the limited-program-supervision setting.

We see a clear advantage of IL in learning the correct program structure in Figure 7, which reports program accuracy at various levels of granularity. We notice that this increase in the program accuracy correlates with an improved generalization performance in Figure 8, even though the training accuracy of all the iterated models is similar. These preliminary experiments on GQA indicate that a learning bottleneck can be beneficial even in larger non-synthetic datasets, but may require additional considerations such as partial resetting to make the training time tractable.

