# OpenReview forum: "Iterated learning for emergent systematicity in VQA"
_ICLR.cc/2021/Conference — ICLR 2021 Oral_

### Official Review · AnonReviewer4 · 2020-10-28
**Interesting and impressive application of iterated learning to VQA**

**Rating:** 8
**Confidence:** 3

**Review:**

The authors apply iterated learning - a procedure originating in CogSci analyses of how human languages might develop - to the training of neural module networks. The goal is for iterated learning to encourage these networks to develop compositional structures that support systematic generalization without requiring explicit pressures for compositional structures (in past work, such explicit pressures have generally been necessary). The proposed approach brings substantial improvements in systematic generalization across two datasets, SHAPES and CLEVR.

Strengths:

1. The approach is well-motivated, including impressive coverage of prior literature in both ML and CogSci.

2. The approach brings impressive gains in an area that is one of the major weaknesses of current ML systems, namely systematic generalization.

3. In addition to the gains in accuracy, one particularly impressive benefit of this approach is the decreased amount of supervision that it requires compared to past approaches.

4. The paper is generally well-written and easy to follow.

Weaknesses:

1. One of the motivations is to expand the use of iterated learning beyond toy datasets. While SHAPES and CLEVR may be not as toy-ish as datasets used in the past, they still are pretty toy-ish, so I’m not sure if this paper can reasonably claim that one of its contributions is to expand iterated learning to realistic domains.

2. Though the paper in general was very clear, I found Section 3.2 to be a bit hard to follow, and that section is important as it is the part that describes the structure of the iterated learning framing. I think this section would benefit from starting each subpart with a more high-level, intuitive description of what that stage accomplished, before diving into the details.

Minor comments:

1. Fodor et al only has 2 authors - Fodor and Pylyshyn

2. Page 3: “Although, the Gumbel straight-through estimator”: this use of “although” is usually frowned upon - better to use “However”

3. Page 5: typo: “minimzing”

4. In general, for the bibliography, check to see if a paper has been published at a conference or journal; if so, cite that version instead of the arXiv version. E.g., “Neural machine translation by jointly learning to align and translate.” was published in ICLR 2015, and “Systematic generalization: what is required and can it be learned?” was published at ICLR 2019.

---

> ### Author Response · Authors · 2020-11-16
> **Response to AnonReviewer4**
>
> We thank the reviewer for taking the time to read our paper and for their valuable suggestions. We have incorporated the reviewer’s minor comments and will make Section 3.2 easier to follow in the paper.
>
> While we agree that SHAPES is fairly toy-ish (244 unique questions in 12 question templates with 30x30 images laid out in a 3x3 grid), CLEVR is substantially more complex in comparison (700,000 unique questions in 90 question templates, with 3D-rendered 224x224 images). As a crude point of comparison, it takes about 30 minutes to train a SHAPES-SyGeT model to achieve decent generalization on a modern GPU, whereas it takes over two days to do the same on CLEVR. That said, CLEVR is indeed a synthetic dataset, although this has the advantage of allowing us to clearly demonstrate the systematic generalization gains of our method by removing other sources of error.
>
> As we mention in the separate standalone comment, we understand the need to test methods such as ours on realistic data and are thus experimenting with the GQA dataset. These results are not necessary for the goal of our paper, and we will be unable to explicitly evaluate systematic generalization as we do in the case of SHAPES-SyGeT and CLEVR/CLOSURE. Thus, we plan to include the validation performances in our appendix rather than the paper’s main body, which we will do before the camera-ready deadline.
>
> Due to the subjective definition of “toy” tasks, we can modify that claim in the paper to specify that we open up IL to broader machine learning applications beyond the previously-explored scope of language emergence and preservation.

---

> > ### Comment · AnonReviewer4 · 2020-11-23
> > **Thank you for the response!**
> >
> > Thank you for the responses! I think it would be nice for the paper to use different language than "toy", purely because I worry that some readers will view the paper negatively on the grounds that SHAPES and CLEVR still are toy-ish (due to the subjective nature of the definition of "toy", as you noted). So I would recommend using different terms that are more objective, to prevent readers from misreading the paper in that way. (E.g., you could say that you have applied IL to datasets that are substantially more complex than those used in prior work - "substantially more complex" seems a fair statement to me, as it makes not claim that SHAPES and CLEVR are not toy datasets).
> >
> > By the way, I very much support the use of synthetic datasets, as they are more controlled and therefore allow for more rigorous experiments. The only concern I had was that the current phrasing might make the paper seem like it is overclaiming.
> >
> > Anyway, this is a small enough issue that it has not affected the score I assigned to the paper.

---

### Official Review · AnonReviewer1 · 2020-10-30

**Rating:** 7
**Confidence:** 3

**Review:**

This paper proposes to combine iterated learning (the process of repeated language transmission from a ‘parent’ agent to a ‘child’ agent) with neural module networks (NMNs), in order to emerge NMN layouts that perform better at systematic generalization. The paper evaluates on their new variant of the SHAPES dataset that tests systematic generalization, and on CLEVR / CLOSURE, showing improved systematic generalization performance while requiring a small amount of ground-truth layout supervision.

Pros:
- I think the idea of combining IL with NMNs is really clever (heh). Treating the program generator and execution engine as two agents that need to coordinate through a shared language (NMN layouts) is really interesting. If it were to work without layout supervision, it could open up new doors to applying IL + NMNs to many other tasks

- The paper is quite well-written, and easy to follow

- The related work section is thorough

- The experiments on SHAPES-SyGeT, show that IL helps significantly for generalization of NMN models

- I appreciate the ablations in the Appendix.


Cons:
- One of the main questions I have about this approach is whether it will provide any benefit on more complex problems (e.g. large-scale VQA). There are a few reasons to think it might not be able to do so:
1) As alluded to in the paper, the IL procedure requires a lot of compute, which could be used to train larger models on more pre-training data
2) The improvement in validation performance on the more complex CLEVR dataset is fairly modest (though the program accuracy increase is large). While CLEVR is more complex than SHAPES, it is still a fairly artificial dataset targeted ‘compositional’ in nature, compared to general VQA. This suggests that it might be hard to get IL+NMN to work well on harder problems.
3) The method still requires (a small amount of) ground-truth layout supervision, which is not obtainable in general VQA or most other tasks.

- I would also like to have seen a bit more analysis / description of why the method currently fails without any ground-truth layout supervision. I think this would improve the paper a lot, as it would help other researchers improve upon the method to address this problem.



Overall:
I think the ideas in this paper are interesting enough, and the execution good enough, to warrant acceptance. While I have some concerns about whether this approach will scale, these questions will have to be answered in subsequent works and with further research.


Small typos:
“each new-born child need” -> needs

“Recently machine learning community also show” -> the machine learning community also shows

“Minimzing” -> minimizing

---

> ### Author Response · Authors · 2020-11-15
> **Response to AnonReviewer1**
>
> We thank the reviewer for taking the time to read our paper and for their comments.
>
> - Compute requirements of iterated learning (IL):
> The compute required for IL depends on the compute required to train the program generator (PG) and the execution engine (EE) independently and the frequency of re-initialization of these components. For CLEVR, the EE takes about a day to achieve high validation performance on a modern GPU, even with the ground-truth layouts. Thus, re-initializing the EE for every iteration becomes computationally intractable. Fortunately, the seq2seq PG is quick to train, and thus not reinitializing the EE drastically reduces the computational overhead of IL. With this setup for CLEVR, we find little difference in the compute between IL and non-IL models. This is because IL model training is dominated by the interacting phase, while non-IL models are essentially trained through one long interacting phase.
> However, if the question and program languages are more complex and the PG is a larger model such as a transformer, re-training the PG can become a bottleneck. Making IL work with a harder-to-train PG without completely re-initializing it can be an interesting research direction. Seeded IL ([1]) is one possible strategy to avoid learning the PG from scratch at every iteration. Another idea could be using pruning or strong weight decay on the previous PG to produce an initialization point for the new PG.
>
> - Results for CLEVR/CLOSURE:
> As noted in a separate standalone comment, we have updated our CLEVR/CLOSURE results indicating stronger systematic generalization performance. Regarding in-distribution performance, the improvement on the CLEVR validation set is substantial for Tensor-NMN, which is the widely used NMN architecture from [2]. There is also a modest improvement in the case of Vector-NMN, but baseline performance on CLEVR validation here is already very high.
>
> - Artificial nature of SHAPES-SyGeT and CLEVR/CLOSURE:
> Please see the standalone comment.
>
> - Requirement of ground-truth programs:
> Our method's requirement for ground-truth programs primarily arises from the optimization difficulties in jointly training the PG and the EE from scratch. The EE modules can specialize in performing specific roles only if the programs reuse the modules in appropriate ways. On the other hand, a good quality reinforcement signal from the EE to the PG relies on some amount of specialization of the modules. However, we note that only 100 programs is sufficient for us to achieve CLEVR validation performance close to that of previous NMN papers, which used 18000 ([2]) or 1000 ([3]) ground-truth programs. With a reduced need for supervision, it might become tractable for practitioners to achieve good systematic generalization in realistic VQA tasks by labeling programs for only a small set of questions.
>
> References:
> [1] Yuchen Lu, Soumye Singhal, Florian Strub, Olivier Pietquin, and Aaron Courville. Countering language drift with seeded iterated learning. arXiv preprint arXiv:2003.12694, 2020.
> [2] Justin Johnson, Bharath Hariharan, Laurens Van Der Maaten, Judy Hoffman, Li Fei-Fei, C Lawrence Zitnick, and Ross Girshick. Inferring and executing programs for visual reasoning. In Proceedings of the IEEE International Conference on Computer Vision, pp. 2989–2998, 2017.
> [3] Ramakrishna Vedantam, Karan Desai, Stefan Lee, Marcus Rohrbach, Dhruv Batra, and Devi Parikh. Probabilistic neural-symbolic models for interpretable visual question answering. arXiv preprint arXiv:1902.07864, 2019.

---

### Official Review · AnonReviewer5 · 2020-11-07
**Revised review**

**Rating:** 6
**Confidence:** 4

**Review:**

Review:
The authors address methods to encourage the emergence of the layout expression structures on the frameworks of neural module networks (NMN) for the visual QA problems. The methods are motivated from the works on language emergence for communication between multi-agents and the language acquisition of new-born babies from parents, which achieved with limited data. The methods, ‘iterative learning’ (IL) are designed as forming two agents (program generators and execution engines) to play VQA games. Basic architectures and learning methods seem to be very similar to the approach of semi-supervised learning introduced in [ICCV17].
This paper deals with one of very interesting topics, the language emergence among cooperative multi-agent environments and the compositionality of human language as recent related studies are well-surveyed in related work. In particular, the main idea of problem formation, layout expressions in NMN as emergent languages is very fresh and interesting.
The main claims are as follows: (1) the proposed approach of IL improves generalization performance for visual QA, and it is shown experimentally by comparing the ablation results of IL. (2) the language structures in the ground-truth data are recovered with only limited supervisions and the superiority is validated on two datasets – SHAPES-SyGeT and CLOSURE. However, I believe that the evidences for their claims are insufficient. Specifically, the authors do not provide enough information of language structure such as the superiority compared to other methods and the structure similarity of recovery levels.
I recommend 'ok, but not good enough – reject’ for this paper.


Pros:
- The authors propose novel interesting problem and their solutions. Arguably, it seems potentially to be on one of important research flows to make influence to lots of works for academia in the future.
- They find and report good performance for out-of-distribution accuracies for visual QA datasets.

Concerns:
- It is not clear which parts in the proposed methods are novel with respect to previous works. Those make vague which parts are the authors’ contributions.
- I think IL should be clarified from semi-supervised learning approaches on them for visual QA. Also, for reproducibility, it should be specified which parts are with/without IL. What is ‘learning bottleneck’ on this approach? Also, it is not enough how program generators and execution engines are specified, even though some explanations are in appendix. I think it needs more links for reference or explanation.
- As mentioned above, the supports for main claims are not appropriate or unclear. It needs to theoretically or experimentally show the results of comparison with other methods and the similarity of the recovery level of language structures.
- Table 1 reports the result of comparative methods such as MAC and FiLM without program supervision. How are they configured in the experiments?
- I think that it would be better understandable to show usability and superiority with the experimental results on realistic visual QA such as VQA and GQA.

Minors:
- In Section 3.1,  γ. -> γ,  β. -> β
- It needs the description for operators and symbols for the formulae in Section 3.1

[ICCV17] Johnson et al., Inferring and executing programs for visual reasoning, ICCV 2017.

------------------------------------------------------------------------------------------------------------------------------
After rebuttal

From the revised version of this manuscript, the authors resolve my major concerns such as clarity/reproducibility of the method, differentiation from the previous works including semi-supervised learning, and scalability. So, I've raised my score to 6.
Thank you for the contributions!

---

> ### Author Response · Authors · 2020-11-14
> **Response to AnonReviewer5**
>
> We thank the reviewer for their insightful comments.
>
> - Contributions and note on other datasets:
> We have provided a clear set of contributions of this paper, as well as a note on datasets, as a standalone comment. We kindly ask the reviewer to please refer to that comment first.
>
> - Relation to semi-supervised approaches:
> We see iterated learning (IL) as more of a regularization technique than a semi-supervised learning method. IL in general can also be used in unsupervised settings. We happen to apply it in a semi-supervised learning setting for VQA, diverging from its previous scope of language emergence in the machine learning literature. That said, IL is indeed related to semi-supervised learning. Where semi-supervised learning deals with grounding a subset of its predictions in ground-truth labels, IL is supervised using a subset of the previous generation's utterances. The crucial difference is that in IL, the supervision set evolves, changing the language acquired through the learning bottleneck in every new generation until convergence.
>
> - Learning bottleneck:
> In our work, the learning bottleneck primarily arises from limiting the length of the learning phase. In the learning phase, a new program generator (PG) is trained on the previous PG's utterances for a limited number of gradient updates. This is related to early stopping, which is an effective strategy for regularization in machine learning. The central hypothesis for the success of this bottleneck is that structured language is easier to fit for neural networks ([4], [5]). Thus, easy-to-learn structure is acquired quickly by training the new PG for a limited number of steps. In contrast, idiomatic concepts that are harder to acquire (requiring additional learning time or resources) are not transmitted. Consequently, a student PG can learn a more structured language than its teacher. The repeated application of this bottleneck can push the language of programs towards one that is more structured. The ability to solve the task using programs exhibiting structure promotes systematic generalization. If the learning phase's length is too small or too large, the effectiveness of the learning bottleneck can decrease due to underfitting or overfitting the teacher. We will clarify this intuition in the paper.
>
> - Specifics of architecture:
> Since our work's focus is on the IL algorithm, much of our architectural setup is based on previous works ([3] for Tensor-NMN, [2] for Vector-NMN and the PG), and we provide details for our Tensor-FiLM-NMN module architecture in Section 3.1. We will expand Section 3.1 to explain the notations and operational details better, as well as add more details for reproducibility in the appendix.
>
> - Clarity of which parts are with/without IL:
> The details of which parts of the architecture are modified when running experiments with and without IL are specified in Section 3.2. However, for clarity and ease of reference, we will add the algorithm (abstract pseudocode format) describing all the steps of our approach more precisely in the appendix.
>
> - Comparison to other methods and FiLM/MAC setting:
> Since this work aims to demonstrate the effectiveness of IL for encouraging systematic generalization in NMNs, the comparisons of interest are the ones between NMNs with IL and without IL, which we have presented in Section 5. We present the results on FiLM and MAC, which achieve near-perfect accuracy on large datasets like CLEVR (see [6], [7]), not to claim state-of-the-art performance, but to provide a better understanding for readers as to where models without program supervision stand. We know from [1] and [2] that NMNs can systematically generalize better than models with generic deep architectures. However, there are stringent supervision requirements for this to happen in practice, which we address in this work. As with all our models, both FiLM and MAC are configured by finding the hyperparameters that perform the best on Val-IID.
>
> [continued in the next comment]

---

> > ### Author Response · Authors · 2020-11-14
> > **Response to AnonReviewer5 (cont.)**
> >
> > - Recovery level of language structures:
> > Our understanding of the reviewer's concern regarding the "recovery level of language structures" is that they would like to see some measure of how much of the structure is recovered by our methods and baselines beyond just the program accuracy. We would appreciate it if the reviewer could clarify this. However, we respond here with our current understanding to argue that the program accuracy indeed largely gets to what we want for this work.
> > In the context of NMNs, the program determines the computation to be performed on the input images. Small differences in the serialized representation of the computation graph, which the PG outputs, can result in trees that differ significantly in structure. Metrics that use n-gram statistics like the BLEU score can thus be misleading since they may provide high scores for programs that have similar serialized structure but vastly different tree representations once parsed. Since we have some amount of program supervision, there indeed is a notion of a "correct" program structure, which is one that agrees with the supervision programs. Finally, we note a very significant and clear difference in the program accuracy exhibited by IL and baselines without IL that correlates with the models' ability to systematically generalize, which supports our choice of discussing the recovery of structure in terms of program accuracy.
> >
> > - Theory:
> > Finally, while we do believe that a theoretical understanding of IL is necessary, we consider it a broad research problem that is out of scope for our study. A mathematical model from cognitive science is presented in [8], that relies on a Bayesian learner assumption. However, this theory does not sufficiently explain IL in deep learning or in the presence of a grounding task. Our paper offers contributions to encourage the research community to study IL in more detail, which will hopefully offer further theoretical insights.
> >
> > References:
> > [1] Dzmitry Bahdanau, Shikhar Murty, Michael Noukhovitch, Thien Huu Nguyen, Harm de Vries, and Aaron Courville. Systematic generalization: What is required and can it be learned? In International Conference on Learning Representations, 2019.
> > [2] Dzmitry Bahdanau, Harm de Vries, Timothy J O’Donnell, Shikhar Murty, Philippe Beaudoin, Yoshua Bengio, and Aaron Courville. Closure: Assessing systematic generalization of clevr models. arXiv preprint arXiv:1912.05783, 2019.
> > [3] Justin Johnson, Bharath Hariharan, Laurens Van Der Maaten, Judy Hoffman, Li Fei-Fei, C Lawrence Zitnick, and Ross Girshick. Inferring and executing programs for visual reasoning. In Proceedings of the IEEE International Conference on Computer Vision, pp. 2989–2998, 2017.
> > [4] Fushan Li and Michael Bowling. Ease-of-teaching and language structure from emergent communication. In Advances in Neural Information Processing Systems, pp. 15851–15861, 2019.
> > [5] Yi Ren, Shangmin Guo, Matthieu Labeau, Shay B. Cohen, and Simon Kirby. Compositional languages emerge in a neural iterated learning model. In International Conference on Learning Representations, 2020.
> > [6] Ethan Perez, Florian Strub, Harm De Vries, Vincent Dumoulin, and Aaron Courville. Film: Visual reasoning with a general conditioning layer. In Thirty-Second AAAI Conference on Artificial Intelligence, 2018.
> > [7] Drew Arad Hudson and Christopher D. Manning. Compositional attention networks for machine reasoning. In International Conference on Learning Representations, 2018.
> > [8] Thomas L. Griffiths and Michael L. Kalish. Language evolution by iterated learning with Bayesian agents. In Cognitive science 31.3, 2007.

---

> > > ### Comment · AnonReviewer5 · 2020-11-22
> > > **Reply to the Response**
> > >
> > > Thank you for your detailed response.
> > > It is helpful to understand your manuscript more, and it is nice to be ready with the result of realistic VQA datasets.
> > > By the response, some of my concerns are resolved.
> > >
> > > However, there are still unclear parts remained as follows:
> > > (1) on the learning bottleneck: I'm still not sure that how does *the learning bottleneck* give structural constraints or sampling constraints on the training processes.  What is the meaning of limiting the length of learning phase? Is it for masking the training samples similar to [5]? Why is it helpful to improve the proposed method? So, how many iterations on T_p and T_e  are executed finally? 2000 (in Appendices)?
> > >
> > > (2) clarity of which parts are with/without IL: For without IL configuration, which components in IL are not performed? Most experiments include with/without IL cases along the x-axis of "step". It seems that the configuration of without IL also follows 3 phases of iterative learning. Is the learning bottleneck just blocked? It is not still clear.

---

> > > > ### Author Response · Authors · 2020-11-23
> > > > **Response to AnonReviewer5's reply**
> > > >
> > > > We thank the reviewer for their response and thoughtful questions. For the convenience of the reviewer and other readers, we start by giving a short overview of our method. We will refer to this overview later when addressing the reviewer’s specific concerns.
> > > >
> > > > Our iterated learning (IL) approach involves the following phases:
> > > > 1. *Interacting phase:* This is a standard self-play setting where the program generator (PG) and the execution engine (EE) are trained end-to-end to solve the VQA classification task. We use REINFORCE to estimate the gradients of the PG, and also provide direct supervision to it with a small subset of labeled programs. This phase is carried out for a fixed number of gradient updates specified by $T_i$. *In our experiments without IL, this is the only phase executed.*
> > > > 2. *Transmitting phase:* We collect samples from the PG to serve as targets for an imitation learning task in the next phase. We sample training questions $q$ and do a forward pass through the PG to produce predicted programs $\hat{z}$. We add tuples $(q, \hat{z})$ to the transmitting dataset $D$ to contain $T_t$ samples.
> > > > 3. *PG learning phase:* A new PG is initialized and trained to imitate the previous PG, using the transmitting dataset $D$. The new PG is trained to match $\hat{z}$ when given $q$ as input, where $(q, \hat{z})$ is sampled from $D$. This phase is carried out for a fixed number of gradient updates specified by $T_p$. *The value of $T_p$ controls the learning bottleneck by altering the new PG’s tendency to underfit or overfit the previous PG’s language.*
> > > > 4. *EE learning phase:* We initialize a new EE and train it to adapt to the new PG. We train the new EE to solve the VQA classification task using programs generated by the new PG, without modifying the new PG. Adapting the EE stabilizes training during the interacting phase. This phase is carried out for a fixed number of gradient updates specified by $T_e$.
> > > >
> > > > We now address the reviewer's specific concerns.
> > > >
> > > > - **Learning bottleneck**:
> > > > Iterated learning creates a new PG and a new EE for every generation, and a learning bottleneck controls the amount of information that passes from one generation to the next. We impose this learning bottleneck only by limiting the number of gradient updates $T_p$ performed during the PG learning phase. Based on the hypothesis that structured language is easier to fit for neural networks ([4], [5]), we expect systematic global linguistic rules to be acquired before any specific idiosyncrasies or non-compositional rules. A well-tuned $T_p$ would ideally stop the PG learning phase just before any idiosyncrasies are learned (for example, see Table 1 in [5]). Thus, the goal of the learning bottleneck is to allow structured rules to survive to the new generation, while eliminating non-compositional idiomatic rules from the layout language.
> > > > By repeatedly applying this learning bottleneck, our IL algorithm pushes the language towards one that can solve the task while maintaining a structure that is easy-to-learn. It is this combination that aids systematic generalization. *Our algorithm therefore explores regularization that emerges from modifications to the learning process rather than structural constraints.*
> > > > As we specify in Table 3 in the appendix, $T_p$ is 2000 in our experiments. If we initialize the new EE in the EE learning phase from scratch, $T_e$ is 250; if it is initialized as the EE after the previous EE learning phase, $T_e$ is 200; and if it is initialized as the EE after the previous interacting phase, $T_e$ is 50. We find $T_p$ to be the more important hyperparameter since it controls the learning bottleneck, while $T_e$ primarily affects the stability of training.
> > > >
> > > > - **Parts with/without IL**:
> > > > In our runs without IL, the model is trained entirely in one long interacting phase, roughly analogous to previous works that have used NMNs on these tasks. In this case, there are no transmitting or learning phases, and the PG and the EE are never reset.
> > > > In our plots, the x-axis denotes the number of gradient steps. This is straightforward to track in the runs without IL. In the case of IL, we maintain a global counter for gradient steps across all phases. This ensures that our plots fairly reflect the computational requirements of IL. The training task and program accuracies are measured only during the interacting phases, so one can see dips in performance between the end of one interacting phase and the start of the next interacting phase. Validation accuracies are measured at the end of interacting phases in the case of IL, and at regular intervals for the baselines without IL.
> > > >
> > > > We will upload a revision of the paper shortly, updated to address some of the reviewer's concerns. As a reminder, we will not have the GQA results until the camera-ready version since these experiments are time-consuming and not necessary to convey the core message of our paper.

---

> > > > ### Author Response · Authors · 2020-11-23
> > > > **Clarification on the "iterated" part of iterated learning**
> > > >
> > > > We realize that due to the lack of a formal algorithm in our paper that there could be a misunderstanding regarding what is "iterated" in iterated learning (IL). Just in case, we clarify it here.
> > > >
> > > > Our IL approach follows four phases *in a single iteration*, and these phases are repeated many times during training. For brevity, we denote the interacting phase as P1, transmitting phase as P2, PG learning phase as P3, and EE learning phase as P4. Then, the **training of an IL model** goes as follows:
> > > > P1 -> P2 -> P3 -> P4 -> P1 -> P2 -> P3 -> P4 -> P1 -> ...
> > > >
> > > > In contrast, the **training of a model without IL** proceeds through one long interacting phase. Viewed another way, it can be seen as:
> > > > P1 -> P1 -> P1 -> ...
> > > >
> > > > Every time a PG learning phase is executed, a new program generator (PG) is initialized to represent a new generation. Similarly, a new execution engine (EE) is created in EE learning phases (it can either be initialized from scratch or be a clone of a previous EE).
> > > >
> > > > We will add a formal algorithm describing our approach in the appendix.

---

### Author Response · Authors · 2020-11-14
**Our contributions, CLEVR update, and a note on datasets choice**

We highlight our main contributions here and will update the paper to reflect these more clearly. Additionally, we provide a note on some updated results and remark on the choice of SHAPES-SyGeT and CLEVR/CLOSURE for our work.

- Main contributions:
  1. We present iterated learning (IL) as a more fundamental way of recovering structure in machine learning. To the best of our knowledge, IL has not been applied outside language emergence or preservation. We believe this is an important message to share with the machine learning community, where there is still a need and a lack of understanding of the emergence of compositional structure.
  2. We propose an IL method to achieve higher systematic generalization (generalization to novel combinations of known concepts) in neural module networks (NMNs). Previously, NMNs have been shown to exhibit superior systematic generalization ([1], [2]) but only with gold-standard layouts. Subsequently, prior work has required supervision with a large number of ground-truth programs for CLEVR (18000 for [3], 1000 for [4]) to get NMNs to generalize well. In contrast, our method is significantly more data-efficient (only 100 ground-truth programs for supervision) by using IL to learn the program language.
  3. We introduce the SHAPES-SyGeT dataset as a new split of the existing SHAPES dataset to evaluate systematic generalization as a lightweight alternative to CLEVR/CLOSURE.
  4. Minor contribution: Tensor-FiLM-NMN module architecture to combine the parameter-efficient Vector-NMN with the spatial representational power of Tensor-NMN.

- Updates to CLEVR results:
While we already presented clear advantages in recovering the correct programs, the CLEVR models in the first version of our paper did not completely converge. Thus, we have updated the paper with experiments that run twice as long and with 8 seeds instead of 3. With this setup, IL outperforms the baselines on all but one CLOSURE category for both Tensor-NMN and Vector-NMN, illustrating the superior systematic generalization of IL.

- Note on our choice of datasets:
We use SHAPES-SyGeT and CLEVR/CLOSURE for this work to clearly demonstrate the advantages of IL for systematic generalization. These diagnostic VQA datasets make it easy to analyze and split questions into templates, which can then be divided into training and testing templates. We train on questions generated from the train templates and evaluate on questions generated from templates never seen during training (SHAPES-SyGeT's Val-OOD and CLOSURE).
However, we agree with the reviewers that it is important to understand the scaling limitations of methods such as ours. Although not necessary for our paper's goals, we are trying our method on the GQA dataset ([5]) to analyze the gains of our method against the baselines on a large-scale non-synthetic VQA dataset. This dataset also provides programs for its questions, similar to the datasets we have used. We plan to include these results in the appendix before the camera-ready deadline.

References:
[1] Dzmitry Bahdanau, Shikhar Murty, Michael Noukhovitch, Thien Huu Nguyen, Harm de Vries, and Aaron Courville. Systematic generalization: What is required and can it be learned? In International Conference on Learning Representations, 2019.
[2] Dzmitry Bahdanau, Harm de Vries, Timothy J O’Donnell, Shikhar Murty, Philippe Beaudoin, Yoshua Bengio, and Aaron Courville. Closure: Assessing systematic generalization of clevr models. arXiv preprint arXiv:1912.05783, 2019.
[3] Justin Johnson, Bharath Hariharan, Laurens Van Der Maaten, Judy Hoffman, Li Fei-Fei, C Lawrence Zitnick, and Ross Girshick. Inferring and executing programs for visual reasoning. In Proceedings of the IEEE International Conference on Computer Vision, pp. 2989–2998, 2017.
[4] Ramakrishna Vedantam, Karan Desai, Stefan Lee, Marcus Rohrbach, Dhruv Batra, and Devi Parikh. Probabilistic neural-symbolic models for interpretable visual question answering. arXiv preprint arXiv:1902.07864, 2019.
[5] Drew A. Hudson and Christopher D. Manning. Gqa: A new dataset for real-world visual reasoning and compositional question answering. Proceedings of the IEEE Conference on Computer Vision and Pattern Recognition, 2019.

---

### Decision · Program_Chairs · 2021-01-07
**Final Decision**

**Decision:**

Accept (Oral)

**Comment:**

This paper presents an original perspective on how to learn layouts and modules of neural module networks jointly, in a way that encourages the emergence of compositional solutions. In particular, layouts are treated as messages from an emergent language, and iterated learning is used to encourage the emergence of structure. The paper shows good performance in inducing compositional structure in two datasets.

Summarizing the reviewers' doubts, one is that the idea is tested on relatively toyish data sets, and it is not clear how it would scale up. The second, coming from one reviewer, concerns a lack of originality that, honestly, I do not see. If anything, this is probably the most original paper in my pool.

Concerning the first point, that is a fair objection, but I think that getting good results on program learning on datasets such as CLEVER is more than encouraging for a paper that is introducing quite a novel idea for the first time.

Finally, the authors added new text and new experiments strenghtening their conclusion during the discussion.

I am strongly in favour of accepting this paper.